# OpenCare5G: O-RAN in Private Network for Digital Health Applications

**DOI:** 10.3390/s23021047

**Published:** 2023-01-16

**Authors:** Wagner de Oliveira, José Olimpio Rodrigues Batista, Tiago Novais, Silvio Toshiyuki Takashima, Leonardo Roccon Stange, Moacyr Martucci, Carlos Eduardo Cugnasca, Graça Bressan

**Affiliations:** 1Department of Computer Engineering and Digital Systems, Escola Politécnica, University of São Paulo, São Paulo 05508-010, Brazil; 2Department of Clients & Industries, Deloitte Touche Tohmatsu, São Paulo 04711-130, Brazil; 3NEC Latin America S.A., São Paulo 05001-100, Brazil; 4Meta, Menlo Park, CA 94025, USA

**Keywords:** 5G, B5G, digital health, Open RAN, private network

## Abstract

Digital Health is a new way for medicine to work together with computer engineering and ICT to carry out tests and obtain reliable information about the health status of citizens in the most remote places in Brazil in near-real time, applying new technologies and digital tools in the process. InovaHC is the technological innovation core of the Clinics Hospital of the Faculty of Medicine of the University of São Paulo (HCFMUSP). It is the first national medical institution to seek new opportunities offered by 5G technology and test its application in the first private network for Digital Health in the largest hospital complex in Latin America through the OpenCare5G Project. This project uses an Open RAN concept and network disaggregation with lower costs than the traditional concept used by the telecommunications industry. The technological project connected to the 5G network was divided into two phases for proof-of-concept testing: the first with an initial focus on carrying out examinations with portable ultrasound equipment in different locations at HCFMUSP, and the second focusing on carrying out remote examinations with health professionals in other states of Brazil, who will be working in remote areas in other states with little or no ICT infrastructure together with a doctor analyzing exams in real time at HCFMUSP in São Paulo. The objective of the project is to evaluate the connectivity and capacity of the 5G private network in these the proof-of-concept tests for transmitting the volume of data from remote exams with higher speed and lower latency. We are in the first phase of the proof of concept testing to achieve the expected success. This project is a catalyst for innovation in health, connecting resources and entrepreneurs to generate solutions for the innovation ecosystem of organizations. It is coordinated by Deloitte with the participation of the Escola Politécnica da USP (The School of Engineering—University of São Paulo), Airspan, Itaú Bank, Siemens Healthineers, NEC, Telecom Infra Projet, ABDI and IDB. The use of 5G Open RAN technology in public health is concluded to be of extreme social, economic, and fundamental importance for HCFMUSP, citizens, and the development of health research to promote great positive impacts ranging from attracting investment in the country to improving the quality of patient care.

## 1. Introduction

Digital Health uses information and communication technology (ICT) resources to produce, provide and make available reliable information about the health conditions of patients. It is a strategic inclusion mechanism, making specialized health assistance available to remote places in Brazil [1]. Digital Health incorporates recent advances in health innovation technology by implementing hardware (HW), software (SW), IoT devices, big data and artificial intelligence (AI) [2] to concentrate high volumes of information in databases, applications, and networks. It enables more accurate follow-up and brings greater security to patients, especially when performing tests remotely in near-real time to obtain the fastest diagnoses. Private networks are also a good tool for ensuring that communication can be safe and provide the desired network quality for health professionals and public managers in the healthcare system [3].

Doctors from InovaHC, Center for Technological Innovation at the Hospital of Clinics, Faculty of Medicine, University of São Paulo (HCFMUSP), started an evolutionary 5G technological project to improve the provision of basic services to populations living in remote areas of Brazil. The initiative, called “Aysu” in the Tupi-Guarani language, will bring medical assistance to the riverside and indigenous population of the Amazon Region. In August 2021, interventional radiologists from the Institute of Radiology at HCFMUSP (InRad) landed on the banks of the Tapajos River, Para State, 2348 km from HCFMUSP in the city of São Paulo, to perform ultrasound exams and assist in the medical care of populations in situations of vulnerability to diseases and epidemic.

The doctors at InRad are part of Zoe, a non-governmental organization (NGO) established to offer treatment and care to those without or with little access to qualified healthcare. The doctors spent a few days providing care on the boat Abare, a kind of basic floating health unit on the river that offers primary care, carrying out 200 exams with two pieces of portable ultrasound equipment of the most varied types in children, young people, men, and women.

With the COVID-19 pandemic, the role of the Abare boat in assisting riverside communities became even more important. The vessel is where people receive basic supplies and protection items, such as masks and alcohol gel [3]. HCFMUSP is a large public technological health center with equipment and devices with all the necessary resources in place, but in the remote places served, these resources are minimal. In addition to primary care, the doctors performed cancer and thyroid diagnoses, pregnancy and hernia tests, not to mention the exchange of experiences between doctors and the local population and interaction with the local communities’ indigenous leaders, shamans, and chiefs. Through this exchange, the doctors were able to experience the dynamics of the village and gain access to knowledge about the forest and local wisdom based on the concept of community life.

In the third expedition of the InRad doctors in 2022, they carried out 377 clinical consultations in children, the elderly, and chronic patients, in addition to 125 ultrasound examinations in indigenous communities of Alto do Xingu. The action, carried out in partnership with “SESAI, more Indian Health” and the NGO “Xingu + Catu”. The difference in this expedition was the provision of on-site service to assist the indigenous people in the villages, allowing them to remain in their homes, as shown in Figure 1.

To reach the villages at the base pole in Alto Xingu, in the northeast region of the State of Mato Grosso, in the southern portion of the Brazilian Amazon, the path taken by health professionals is long and they need to travel by ferry, boat, car, and even by foot, but the effort is rewarded by being able to take the portable ultrasound equipment to these communities.

Then InovaHC professionals developed a partnership with the diverse ecosystem of different actors among technology, telecommunications, government, universities and financial institutions. Called the OpenCare5G Project [4], it is a highly innovative initiative to study and test healthcare applications in the construction of the first 5G private network. The Open Radio Access Network (O-RAN) utilizes the O-RAN 5G concept [3,5,6,7,8,9,10,11,12] to transport health data and to help in the analysis of medical exams by health professionals using local and remote hospital portable devices and equipment.

With this implementation, health professionals will be able to provide care in the middle of the rain forest using this equipment or others connected to the 5G private network safely, and carry out immediate diagnoses with their peers at HCFMUSP in São Paulo. The project has been coordinated by Deloitte and implemented by Airspan, Escola Politécnica da USP (EPUSP the School of Engineering—University of São Paulo), Itaú Bank, Siemens Healthineers, Telecom Infra Project (TIP), Brazilian Agency for Industrial Development (ABDI), Inter-American Development Bank (IDB) and NEC, responsible for the integration of the O-RAN 5G network [4].

The Itaú Bank, the first bank in Brazil to use O-RAN to provide a 5G connection to its network, will provide its expertise in technology, which will make it possible to carry out pilot projects. Bank experts will explore the possibilities of an innovative system in a joint learning process. An important point is that in the project, a 5G Core Network (5GCN) will be used at Itaú Bank (with Cisco, Dell and HP equipment), using the infrastructure of its datacenter, located in the Ipiranga neighborhood, 9 km from HCFMUSP with a LAN-to-LAN fiber optic connection provided by the local telecom company.

The use of a 5G private network aimed at the “freedom” to make the bank’s indoor network, without depending, in principle, on incumbent telecom companies. Itaú Bank has 6000 datacenters, since each branch is considered one of these centers. That is, there is the potential to have 6000 agencies connected to the private O-RAN network throughout the country.

Field tests will be carried out with “real use cases” of data received and sent from exams, images, and voice to evaluate the data-transmission capacity of 5G connectivity, the response time and the feasibility of performing remote exams using the higher speed and lower latency than the 5G and Beyond 5G (B5G) private networks could offer [13]. The project proof of concept (PoC) will initially focus on the first phase to perform local and remote ultrasound scans, later extending to tests of CT images that will be sent to the cloud, in which there will be processing at the edge through machine learning and AI, collaborating to improve the patient’s journey and provide more access to health services, since the analysis of diagnosis can currently take days.

The remainder of this paper is organized as follows: Section 2 contextualizes the related work; Section 3 presents the proposed solution framework; Section 4 describes the scenario, the methodology and PoCs used in the experiments, and reports and discusses the results. Finally, Section 5 concludes the work.

## 2. Related Work

This section contextualizes some problems that may arise with the application of O-RAN in Digital Health in heterogeneous mobile network scenarios for 5G and the use of 5G-CPE (Wi-Fi 6) to support the expected contribution of this work.

The fundamental shift in networking has resulted in three independent but dominant trends in the tech industry: virtualization, open ecosystems and cloudification [14]. These concepts are not new, but the cell phone industry has been delaying their adoption. Core network virtualization is complete, and all the focus is now on RAN virtualization.

There are no articles dealing with the use of a 5G private network employing O-RAN for Digital Health in Brazil. This is the first article aiming at that.

Table 1 summarizes the related work.

The authors of [15] propose that O-RAN takes advantage of hardware and software disaggregation [16,17,18] and creates a unified architecture with various advancements, such as low latency. In addition to facilitating network automation, O-RAN provides several benefits, including:Agility in software-enabled architecture unification makes network suitable for existing, past and future generations;Deployment flexibility for disaggregation and software association makes the network flexible for installation, update and extension;Real-time responsiveness, since vRAN is service-specific network-oriented software that behaves based on the intended service to real-time services and requires very low latency from less critical services;Reduction in operating costs by estimating the plug-and-play feature of O-RAN; modern learning methods can reduce the maintenance cost dramatically when compared with traditional solutions.

The result is that, by placing the software at the center of the network, it is possible to unify the connectivity gains of all generations under one umbrella. By doing so, millions of dollars may be saved [15].

O-RAN was developed for democratizing access to and reducing the cost of future mobile data networks by supporting network services with various Quality of Service (QoS) requirements, such as massive IoT or IoT devices (e.g., portable ultrasound equipment) and uRLLC [19]. In O-RAN, networking functionality is disaggregated into radio units (RUs), distributed units (DUs) and centralized units (CUs), which allows for flexible software in COTS. Furthermore, mapping RU shift requirements to local mobile edge computing centers for future centralized processing would significantly reduce power consumption in cellular networks.

That is, the study of the resource allocation problem between the RU-DU of the O-RAN system was modeled as a 2D packing problem of RU-DU resources in O-RAN, proposing a self-play reinforcement learning strategy. They apply the combined approach of deep neural networks with Monte-Carlo Tree Search (MCTS) to solve this combinatorial optimization problem.

With high throughput and function approximation capability, a deep neural network modeled resource allocation policy can generalize under dynamic network conditions. Learning by auto-run eliminates the need for demonstrator data, which can be expensive and time-consuming to collect [19].

The authors describe that the application of ML as a data analysis method allows machines to explore data and make predictive and proactive decisions in real time [20]. ML can play significant roles in learning from variations in the wireless environment, categorizing problems, anticipating challenges, predicting outcomes, and exploring possible solutions, decisions, and actions [20].

The ML framework can exploit data from different types of UEs to predict traffic volume and dynamically allocate available network resources. In 5G, the RRM (Radio Resource Management) framework in the RAN includes various control functions based on radio measurements and other observations of various user devices or network elements.

Current RANs are reactive and base stations run algorithms on CUs to meet user demands. However, in 5G, even milliseconds of delay can cause a great impact. To enable certain mission-critical applications, such as remote robotic surgery, the 5G network must be predictive and proactive, rather than just reactive. In O-RAN, computing and storage resources must be distributed among the different datacenters that can host the ML algorithm to proactively serve users.

Sustaining advanced Radio Link Monitoring (RLM) requires an efficient implementation of ML-based traffic optimization that can handle large volumes of data in 5G networks. This learning framework would be able to run algorithms autonomously to deal with latency, meeting the RRM functionality and reliability requirements of users [19].

Polese et al. [21] define the management and optimization of new network systems that require solutions that, by opening the RAN access network, expose data and analysis and allow advanced data-driven optimization, closed-loop control and automation. They describe that the challenges are not easy to overcome, as current approaches to cellular networks are the opposite of open. In RAN, network components are monolithic units, complete solutions that implement each layer of the cellular protocol stack, provided by a limited number of vendors and seen by operators as black boxes [21].

O-RAN embraces and extends the 3GPP NR 7.2 split to base stations between the RU and the DU. We can thus have the functionalities of the base station in CU, DU and RU. Furthermore, it connects them to intelligent controller systems through open interfaces that can transmit telemetry of the RAN and implement actions and policies to control it.

Note that the O-RAN architecture includes two RAN Intelligent Controllers (RICs) that perform network management and control at near-real-time (10 ms to 1 s) and non-real-time (greater than 1 s) timescales, which have not been used in the OpenCare5G project.

Future work will be focused on extensions to real-time control loops, including loops that operate in the real-time domain, i.e., below 10 ms, for managing radio resources at the RAN node level, or even below 1 ms for device management and optimization. Typical examples of real-time control include programming, beam management and feedback-free detection of physical layer parameters (e.g., modulation and coding scheme, interference recognition). These loops, which have a limited scale in terms of optimized devices, are not part of the current O-RAN architecture, but are mentioned in some specifications as for further study [21].

## 3. Proposed Solution

OpenCare5G has been conducted according to the efforts presented below.

### 3.1. OpenCare5G Project Architecture

The first step was to define a new approach using the Private Health Network Architecture with the trends in the design of a Gigabit-based packet data network, sharing the 5G Internet with the Customer Premises Equipment (CPE) in the local network via Wi-Fi 6 (802.11ax) that imposes new requirements to build, test, operate and update it.

The 5G xHaul O-RAN Operations, Administration, and Maintenance (OAM) Private Network Project Architecture via vRAN was divided into three segments of logical “nodes”, also called functional blocks, that are areas/connection points for the functions. Then an Infrastructure Layer with the Network Fronthaul (gNB-RU) of the Radio unit, Midhaul (gNB-DU) of the Distribution Unit, and Backhaul (gNB-CU) of the virtual centralized unit towards the 5GCN in the Itaú datacenter was provided, forming the access network by virtualized radio (vRAN) of the End-to-End (E2E) design [22].

We have requested of Anatel, the Brazilian telecommunications regulator, the use of the radio frequency spectrum by telecommunications systems associated with the Private Limited Service (PLS) for allocating the 5G band of restricted interest for health [1,3,23]. This is meant for exploration nationwide, in the private regime with a frequency spectrum in band n77 5G 3700–3800 MHz, 100 MHz bandwidth with vRAN performing baseband functions as software [3,24].

The 5G xHaul O-RAN OAM vRAN Private Network Project Architecture will provide connectivity and interoperability through radios, hardware, software, containers, virtual machines, and cloud-native open interfaces from different manufacturers with an approach of using less hardware and achieving greater economy. The ability to increase flexibility and easily scale up and down workloads should scale to meet changing network demands for mobile bandwidth, ultra-reliable secure communication and ultra-low data latency as shown in Figure 2 [14,25,26,27].

Figure 3 shows the 5G xHaul network architecture of the Inovac OpenCare5G Project with Complete O-RAN OAM via vRAN in the infrastructure, mobile and services layers, showing the subnets connected to the User Equipment (UE)/Exam Room, UE/Control Room Doctor, Radio Units/Subnetwork, 5G CPE (Device), Fronthaul Transport Network, Subnetwork/Sync, DU/vDU, Midhaul 1 Transport Network, Midhaul 2 Transport Network, CU/vCU, Backhaul Transport Network, 5GCN in Cloud and Network Slice for 5G NR.

Figure 2 and Figure 3 show the following components:1.The Radio Subnetwork of the gNB-RU (Fronthaul) which has two radio units (RUs) formed by an active internal antenna (Advanced Antenna System - AAS RU) coupled to the Remote Radio Unit (RRU) for the 5G NR Sub-6GHz type 4T4R. The physical interface for glass fiber optics to control coverage and mitigate interference with “digital beamforming” on demand with internal connectivity is able to increase the transfer rate for speeds above 2 Gbps up to 10 Gbps for the two distinct environments (Exam and Control Room) at HCFMUSP with the following specifications [29,30]:Type Airvelocity 2700 RU from Airspan company for 5G RAN;2 AAS-RU units with 4 4T4R AAS active antennas, 3.7–3.8 GHz range, n77 band, FM, DC;5G Sub-6 radiates radio frequency energy;F2 Fronthaul interface between the RU and the DU for 10 GbE;vRAN Split 7.2x; the RU/DU must conform to functional split RAN 7.2a, Duplex Operation Mode and 100 MHz Bandwidth [29];2.The Indoor 5G CPE in the subnetwork of the gNB-RU (Fronthaul) with two CPEs provides connectivity, speed and latency similar to optical fiber connecting electronic devices via 5G; it distributes signal by Wi-Fi 6 (802.11ax) to the doctors’ examination room and ultrasound equipment examination room at HCFMUSP. Its implementation alleviates network congestion and efficiency in the face of high traffic demand;3.The Fronthaul Transport Network of gNB-RU is connected by glass fiber optics to obtain broadband with synchronous ethernet interface, using wave-length division multiplexing (WDM) connected to RU to improve the efficiency/throughput of the medium with an enhanced common public radio dynamic interface (eCPRI PDU) encapsulated in ethernet for VLAN, class 2;4.Between RU/DU, O-RAN Alliance (Open RAN) of User Plan (UP) traffic service points, time synchronization via Global Positioning System (GPS) [25,31]. At this stage we have the Fronthaul packet network carrying the eCPRI PDU traffic so that it is virtually lossless. Depending on the use case in 5G, the latency in the Fronthaul network is not more than tens of microseconds;5.The DU/CU must combine to allow dual connectivity between the Sub-6GHz and 5G network bands, as well as coordinated transmission to eliminate interference to the minimum possible [29];In the “Subnetwork/Sync” we have the switch and the GPS connection with the gNB-RU (Fronthaul);The DU/vDU of the gNB-DU (Midhaul) of the distribution unit (DU) mounted on a COTS server with docker system for vDU containerization;Midhaul 1 Transport Network from gNB-DU (Midhaul) with glass fiber optics and split 2.0 interface for LAN-to-LAN connection Midhaul 2 Transport Network from gNB-DU (Midhaul) to Unit Centralized (5G-O-CU);The CU/vCU of the gNB-CU (Backhaul) of the Centralized Unit (5G-O-CU) mounted on a COTS Server performing the containerization and virtualization of the vCU in the native cloud connected to the 5G Core (5GC) server, which is connected to the datacenter in Itaú, 9 km from HCFMUSP.

The remote unit (RU) hosts the L1 layer and the lower PHY-Low sublayer in real time, and the vCU is responsible for non-real-time functions such as RAN resource management, encryption and retransmission. Normally, a single vCU can manage many DUs, and a single DU can connect to multiple RUs [24].

In 5GCN, the Service Based Architecture (SBA) defines the network components with a set of network functions interconnected by containers and non-containers, as shown in Figure 2 and Figure 3.

#### 5G NR/NGC Protocol Framework Stack

Figure 4 depicts the conceptual model of the open systems interconnection (OSI) being used in 5G, based on the functionalities of each layer and distinguished by the concepts of services, interfaces and protocols. The OSI model divides functions into seven layers that serve as a reference for developers to program networks and so that they can communicate and work independently of the manufacturer. It is a theoretical model for developers and scholars and the great secret is standardization and interoperability [32].

The TCP/IP model is a four-layer oriented model, forming the stack of the protocol structurecurrently in use, but we need to know both models to interpret the 5G Protocol Framework Stack.

With the implementation of the structure of the datacenter LAN/4G/LTE stacks for the 5G NR/NGC, the layers and sublayers of the structured stack of protocols for interconnection, communication and other functionalities of hardware and software underwent alterations and were included in the functional division options proposed by 3GPP in conjunction with disaggregated O-RAN.

Based on the simplified Radio Protocol Stacks in the comparison of UMTS/LTE/5G Air Interface or datacenter LAN with 4G/LTE/5G RAN network stacks, we will have the “5G NR/NGC SA Protocol Stack” with different technical terms and a functional division“Intra-MAC split” among RU/DU/CU, as shown in layers L1, L2 and L3 and their sublayers in Figure 4 [33].

Based on the OSI model, each layer has its own structure called Protocol Data Unit for data, segments, packets, frames and bits (PDU). This division into layers brings several advantages, such as the decomposition of network communications into smaller, simpler parts and standardization of network components, enabling communication between different types of hardware and network software, preventing changes in one layer from affecting the others. Each layer of the model must provide its services exclusively to the layer immediately above and, consequently, the function of each layer depends on the services of the layer immediately below [32].

Therefore, based on the OSI/TCP-IP model, the “5G NR/NGC SA Protocol Structure Stack” was formed, in which 3GPP released the 38.300 v1 specification on NR and NG-RAN. These standards, termed Phase 2, include details about the 5G NR network and the protocol architecture shown in Figure 4 with the three layers L1, L2 and L3 and their sublayers [34,37]:Layer L1 on sublayer RF, PHY Low(L) and High(H);Layer L2 on sublayer MAC (L/H)/(SCH), RLC (L/H) and PDCP (L/H);Layer L3 on sublayer RRC and SDAP, only SA and not NSA, NAS and IP.

Figure 5 demonstrates the architecture of layers L1, L2 and L3 for DownLink (DL) and UpLink (UL), where the:PHY sublayer offers transport channels to the MAC sublayer;MAC sublayer offers logical channels to the RLC sublayer;RLC sublayer offers RLC channels to the PDCP sublayer;PDCP sublayer offers Radio Bearers to the SDAP sublayer;SDAP sublayer provides the QoS flows.

The “5G NR/NGC SA Protocol Framework Stack” is designed to support lower delays and higher data rates with QoS in the sublayers shown in Figure 6 for UE and 5G gNB base station with DL or UL directions of data packets [37].

As shown in Figure 5:Packet Data Convergence Protocol (PDCP) is the first sublayer in the 5G NR protocol stack that receives/transmits network layer traffic (TCP/IP traffic);Data Radio Bearer (DRB) has the logical connection used within the 5G protocol stack to carry protocol data units (PDUs);The Service Data Adaptation Protocol (SDAP) functionality is to map the QoS flows for the DRB in the PDCP sublayer in the DL and UL directions [40].

The channels shown in Figure 5 help us organize and simplify the design of the stack for each sublayer. It is prioritized and optimized in different ways for the channels during the DL and the UL [41,42]. There are three types of specific channels [40]:Logical channels that define the type of data to be transferred with the traffic (user data), such as “paging messages”, dedicated control information;Transport channels that define the information to be transported to the physical layer and the characteristics of the data, such as error protection, channel coding, the cyclic redundancy check (CRC) and the size of the data packet;Physical channels characterized by their time and access protocols, data rates as traffic channels [40].

### 3.2. Fronthaul Challenges and Functional Division Options

O-RAN provides the option of placing functions on the network at different locations and along the signal path; this option is called a functional RAN split and is referred to as Lower Layer Split (LLS) and Higher Layer Split (HLS), as shown in Figure 3. The project’s 5G private network is allowed to improve its performance and make some compensations due to RAN disaggregation, making the decision of which node/site/unit should control certain operations of the functional division of the RAN.

3GPP defined Options 1–8 of interfaces for the functional division to interconnect them, in the F2 network interfaces of the gNB-RU (Fronthaul) to connect (RU->DU), in which we have the eCPRI traffic interface (class 2, functional split 7.2x) in Figure 6. eCPRI requires time synchronization (frequency and phase) to be achieved by the packet-in-band method through IEEE 1588v2 to operate reliably in the project [43]. The F1 interface (functional split 2.0) on the gNB-DU (Midhaul) (DU->CU) was previously discussed and is shown in Figure 6 in relation to layers L1, L2 and L3 of the RU, DU and CU [44].

**Figure 6 sensors-23-01047-f006:**
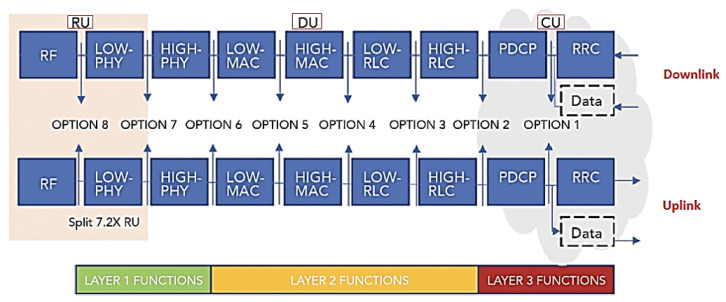
Eight functional split options with 5G xHaul network architecture for gNB-RU (Fronthaul) (RU->DU), gNB-DU (Midhaul) (DU->CU) and Backhaul (CU->core network). Based on [40].

The functional division options allow the Fronthaul and Midhaul network interfaces of the DU/CU to face the challenges of increased traffic and the intelligence of placing close to the RU (AAS-RU and the RRU) versus the DUs [45,46]. O-RAN can distribute the baseband processing over the logical nodes, thus defining the characteristics of these links, choosing which of the RAN’s functional divisions (Split of 2-8), logical to the sublayer of the device, improves the split ratio. Fronthaul to Backhaul (Device->RU->DU->CU [5G NGC]) in relation to the sublayers of the 5G NR/NGC SA is shown in Figure 7.

In OpenCare5Gt, the F2 option was used with the Fronthaul 7-2x functional division between the RU->DU for the 5G Sub-6 GHz spectrum, to obtain more bandwidth and speed by the RU performed in the physical sublayer “ PHY-Low” and “PHY-High” (Split 7 / Split PHY) in Figure 7 [47].

### 3.3. Installation of the RU (AAS-RU and the RRU)

Figure 8 shows the Antenna Radiation Diagram with angular reference of 20° in relation to the horizon for frequencies of 3400–3800 MHz. The RU is installed on the ceiling of the Intervention Center of the InradHC at HCFMUSP (Exam Room–Ultrasound).

### 3.4. DU (Distributed Unit)

The COTs server (HWs/SWs) DU is installed in the InradHC datacenter, shown in Figure 2 and Figure 3 to perform cDU containerization with Docker (platform as a service with operating system level containers to deliver software in container packages isolated from each other), where baseband processing is performed in real time [49].

The RU/cDU process layers from PHY-High to RLC-High are shown in Figure 6 of “Radio Protocol Stack Simplified in the UMTS/LTE/5G Air Interface” [47,49]. With cDU containerization, we will have flexibility of placing processes in different locations in the O-RAN environment, which, in our use case, is in the InradHC datacenter. This logical node includes a subset of the gNB-RU functions (Fronthaul) with the Spit 7.2x functional division and its operation controlled by the CU/vcCU.

The cDU controls the L2 layer, with the MAC-Low sublayer up to the RLC-High, and in Figure 2, Figure 3 and Figure 9 we have the gNB-DU that hosts the upper part of the “PHY-High” sublayer, partially controlled through the CU to the RLC-High, as shown in Figure 6. Figure 9 shows the X2 terminal which is the interface between the O-RAN RU/DU node and the CU/ACP/Kubernetes (K8s) 5GCN.

### 3.5. 5G Core Network (5GCN)

Our 5GCN is detailed in this subsection.

#### 3.5.1. Core Architecture Based on Services and Reference Point

The Core Architecture is SBA and PR based on the 5GC through the 5G base stations shown in the architecture of Figure 2 and Figure 3 to perform the baseband functions using the vRAN in the project with the interfaces that carry out the interactions with the NFs.

Figure 10 illustrates the 5GCN SBA Architecture and Reference Point (PR) with the E2E CP protocol stack and its main interfaces.

In the project, new open interfaces were defined in the SBA and PR core architecture at points “F” and “E1” within the O-RAN node interface between the PHY and RU sublayers with the eCPRI protocol. One node ends in the CP in “C” of Figure 10 and Figure 11 of the gNB-CU, called gNB-CU-CP. The other node that end the user plane “U” of the gNB-CU and is termed gNB-CU-UP. The default open interface between these nodes is specified as “E1”, shown in Figure 10 and Figure 11.

According to Figure 2, Figure 3, Figure 10 and Figure 11, in OpenCare5G the gNB (5G NR Node) is divided into a Distributed Unit (gNB-DU), a Centralized Unit Control Plane (gNB-CU-CP) and a Centralized Unit Data Plane (gNB-CU-DP). Therefore, there is a separation of the control plane (gNB-CU-CP) and the user plane (gNB-CU-UP) that form CUPS in gNB-CU, as per 3GPP [51].

#### 3.5.2. CUPS—Architecture for Control and User Plane Separation

The 5G/GC(Core) SA network was conceived based on the SBA and the PR of Figure 11 for the communication between the nodes with the network functions (NFs) in which the 3GPP defined an application programming interface (API) that works over the Hypertext Transfer Protocol (HTTP) and follows the Representational State Transfer (REST) model [38].

With the SBA NFs for the access and mobility management function (AMF) and the user plane function (UPF), we will have the division of the 5G NR protocol stack through the user plane (UP) and the control plane (CP) forming the 5G NR/NGC SA Protocol Structure Stack’s separation of control plane and users (CUPS) architecture via F1-C, F1-U and E1, with the gNB-DU, splitting the gNB-CU in gNB-CU-CP and gNB-CU-UP [52,54].

Figure 11 and Figure 12 depict the UP architecture that provides user data, while the CP provides connection configuration for mobility and security between the UE and the 5G(gNB) base station [23]. The 5G NR transports information over multiple physical channels. Both gNB-CU-CP and gNB-CU-DP are NF functions for virtualized networks running on the CU server in 5G vRAN with multiple instances and CUPS concept for expansion on demand, in which [52,54]:A gNB can consist of one gNB-CU-CP for several gNB-CU-UPs and one gNB-DU;At the same time, one gNB can consist of one gNB-CU-CP for several gNB-CU-UPs and several gNB-DUs;gNB-CU-CP is connected to gNB-DU by the F1-C interface;gNB-CU-UP is connected to gNB-DU by the F1-U interface;gNB-CU-CP is connected to gNB-CU-UP by the E1 interface;One gNB-DU is connected to only one gNB-CU-CP;One gNB-CU-UP is connected to only one gNB-CU-CP.

The gNB-CU-Control Plane (gNB-CU-CP) consists of:A logical node hosting the RRC;The CP part of the PDCP protocol of the gNB-CU for an gNB;The gNB-CU-CP terminates the E1 interface connected with the gNB-CU-UP and the F1-C interface connected with the gNB-DU [53].

The gNB-CU-User Plane (gNB-CU-UP) consists of:A logical node hosting the UP part of the PDCP protocol of the gNB-CU for an gNB, and the UP part of the PDCP protocol and the SDAP protocol of the gNB-CU;The gNB-CU-UP terminates the E1 interface connected with the gNB-CU-CP and the F1-U interface connected with the gNB-DU [53].

#### 3.5.3. User Plane and Control Plane for UE/Device and gNB Protocol Stack with Network Functions AMF and UPF

The 5G NR/NGC SA protocol stack for the Access Stratum (AS) is divided into CP and UP with the interfaces shown as UE, gNB, AMF and UPF in Figure 12, based on Figure 2 and Figure 3.

The protocol stack has the following data:Device UP (Device/UE) connects to the UPF network function via IP-to-IP;CP is connected via NAS-to-NAS by the AMF network function;The NAS and RRC sublayers are unique to the PC;The message-based IP transport protocol known as SCTP is used between the eNB (Evolved NodeB) and the MME (Mobility Management Entity) to transport the NAS messages;Reliable message transfer between eNB and MME;RRC deals with the general configuration of a cell including the PDCP, RLC, MAC and PHY sublayers [40];The SDAP sublayer deals with the mapping between the QoS flow and the radio bearers;IP (Internet Protocol) packets are mapped to radio bearers according to their QoS requirements;The PDCP sublayer is primarily responsible for IP header compression/decompression, reordering and duplicate detection, encryption/decryption and integrity protection;The RLC sublayer performs error correction through an automatic repeat request (ARQ) mechanism, segmentation/resegmentation of IP packets (header compression) and sequential delivery of data units to upper sublayers;The MAC sublayer is responsible for error correction through the ARQ3 hybrid mechanism (HARQ), uplink and downlink scheduling [40];The PHY sublayer handles encoding/decoding, modulation/demodulation, multi-antenna processing and mapping signals to physical time-frequency resources;The CP is responsible for control signaling for connection configuration, mobility and security;Control signaling originates from a core network or from an RRC sublayer in the gNB;The PHY sublayer is the key component of the NR physical layer technology that undergoes modulation, waveform, multi-antenna, transmission and channel coding [38].

### 3.6. Network Services Based on Adaptive Network Slicing on Open RAN

Network Slicing is an important 5G 3GPP defined resource that allows PLS to support specific use cases and a dedicated set of these network resources and matched service level agreements (SLAs) for each use case. It has been considered in OpenCare5G.

OpenCare5G has not implemented the infrastructure for virtualization of network functions for orchestration (NFVI Orchestration). Support for network slicing is being improved in the reference version for the project in its first adoption of 5GC, SA Release 16. However, these features will only be implemented later.

### 3.7. Virtualization and Containerization for vDU and vCU

Figure 2 and Figure 3 show the implementation of the Airspan O-RANGE Software architecture through to Figure 13 with container network functions (CNF) for vDU using Docker Engine and container and virtualized network functions (vCU) with hypervisor. These will be used in the project to serve the vRAN, to allow a modular, scalable architecture that is flexibile in its implementation.

Figure 13 shows:Block 1, the physical hardware infrastructure of the O-DU and O-CU servers, network and storage in COTS;Block 2, the Host Server for the host operating system (HOS) connected to the network, offering information, resources, services and applications needed by the user and other nodes in the network;Block 3 provides virtual containers with Docker Engine, which is the heart of the Docker System to realize the client–server architecture application installed on the host server [55]. The system uses Container Docker and Machine Virtualization (VM) concepts over the hypervisor. With containers, processing is performed directly on the host server, using the Docker mechanism to make it faster and lighter [55].

In Block 4, we have the container network functions CNF with implementation via POD (Passed Out Drunk) of K8s with Docker, realizing a set of one or more Linux containers in the K8s application.

PODs are composed of a container in the most common use cases or several tightly coupled containers. These containers are grouped into PODs so that resources are shared more efficiently. That is, we will have 1-n application containers (App) on top of the vDU, as the containers are abstractions in the 1-n application layer with Docker (Bins/Libs) packaging the application and its dependencies into the container in a single operating system (OS) with resources shared between containers [55];Each application in the project virtual container runs in its own isolated process controlled by the Docker Engine, as if it were a host operating system. However, it will not virtualize the entire environment on the cDU and cvCU Servers. It will work on top of the n application and its dependencies, creating virtualization only at the host operating system level and not on the entire server, making the project system light and managing to initialize the applications in seconds compared to the traditional VMs. Less RAM, CPU and storage will thus be used to offer lower latency [55].

Five new application protocols were introduced into the Airspan O-RANGE software architecture in the project using vCU-CP and vCU-UP with end-to-end control plane protocols, depicted in Figure 13:F1AP (F1 Application Protocol) is used to relay Packet Data Convergence Protocol (PDCP) CP messages and Radio Resource Control (RRC) messages between the gNB-DU and the gNB-CU-CP;E1AP (E1 Application Protocol) is used to program GTP-U tunnels for the datapath in gNB-CU-DP;XnAP (Xn Application Protocol) is the control protocol used between gNBs to support a variety of O-RAN-related procedures, such as establishing dual connectivity, coordinating Xn-based handovers (duplication), O-RAN data forwarding and paging. It is used in the SCTP sublayer to coordinate gNB-CU-MC functionality (MultiCast functionality within gNB-CU);NG-AP (NG Application Protocol) provides CP signaling between the node via NG-RAN and the access and mobility management function (AMF). The services provided by the NG-AP are divided into UE-associated and non-UE-associated;SCTP (Flow Control Transport Protocol) is similar to TCP in terms of fundamentals, but adds features that compensate for some weaknesses of TCP.

There is a need to implement requirements to provide a base virtual port for Single Root Input/Output Virtualization (SR-IOV) to allow isolating resources via PCI Express between different users on the server O-DU. When these hardware features enable virtual functions, they can be segmented and accessed on K8s PODs to share network resources and secure network traffic [56].

That is, SR-IOV is used when applications running on K8s clusters need high band-width and low latency performance, and when used with VMs, SR-IOV allows applications to bypass the vSwitch, allowing itself so that the VM directly access the device instead of going through the vSwitch to a CP port and a port for UP traffic per instance on the O-DU Server instance [56].

The Intel SR-IOV Network Device Plugin with K8s extends capabilities to address high-performance network I/O by discovering and advertising SR-IOV Network Virtual Functions (VFs) on a K8s host. Thus, SR-IOV is used for CNFs in 5G configuration. Applications can achieve high throughput, different levels of hardware QoS with low latency [56].

### 3.8. cvCU—5GCN Containerized/Virtualized Unit

The O-CU Server-Centralized Unit utilizes cvCU containerization and virtualization where the less-time-sensitive non-real-time packet-processing functions reside and the unit handles all upper-layer functions of the protocol stack of the PDCP [57].

The cvCU also accounts for non-real-time functions at L2 (MAC-Low) and L3 (PDCP-Low) that are performed at the RRC and PDCP layers to perform data packet convergence in conjunction with CUPS of CP and UP [14].

In Figure 14, there is an interface of the E2 nodes, in which the RIC will not be used, only the controls of procedures and functionalities of this E2 node between two CU-C-Plane endpoints and CU-U-Plane close to near-real-time between cDU and cvCU in 5G [14].

VNFs are managed by the NFVI which encompasses all the network hardware and software needed to support and connect these VNFs in a private network in which a VNF is allowed to be instantiated, managed, scaled up or down.

### 3.9. Platform Control with ACP

The Airspan Control Platform (ACP) is a control system architecture software element for RU, vDU and vCU in servers to manage network functions, as shown in Figure 15.

The ACP software is installed and run on the O-CU server, where all containers are monitored and configured from within the ACP. The network configuration needs to reflect the management K8s so that the ACP can communicate with all nodes within the K8s cluster [60].

After installation, the ACP is installed using the SA credentials to create the data-base backend for the web frontend, in which we have the connectivity of the ACP via the Kubernetes Management Network (KMN) for the ACP to control all PODs contained within the worker nodes [60].

With ACP, the project has the flexibility to allow for single or multiple technologies to take control and provide end-to-end management, and to integrate into Operating System and Support (OSS) stacks to automate network management [61].

## 4. Experimental Evaluation

This work adopts an engineering and empirical strategy. It is based on the scientific method since it applies data analysis to the results obtained under the performed procedures.

The execution methods of OpenCare5G adopt a top-down approach, an organized way of developing network projects, with an analysis of each layer of the 5G architecture, starting with the service layer, where the demands of the applications come from, and ending with the infrastructure layer.

The tests for capturing real data for analysis were conducted according to the explanation in this section. The purpose of the project at this stage is examining the use cases in the service modality of the communication protocols of existing and new equipment that will be incorporated into the 5G network, as shown in Figure 2 and Figure 3. We will have multiple applications, including data communication, video and audio signals, voice, text, as well as capture and transmission of telemedicine data.

### 4.1. Definition of the Test Plan

The test plan was divided into two main phases:The first phase is related to the PoC-1 involving an active 5G private network with the architecture set up at the InRad of HCFMUSP in two rooms—the Intervention Sector Room and Ultrasound Exam Room—to use the portable ultrasound (USG) on a patient lying down on a stretcher. He was accompanied by Doctor 1, who performed the USG of the abdomen, and Doctor 2, a radiologist in the central room in front of a monitoring station who analyzed the data captured from the exams, and whose rooms are covered by the 5G network;The second phase is related to PoC-2 involving the same 5G private network remotely connected to the Computational Tomography MRI/CT.

In these use cases, the focus is also on validating technical criteria involving latency and signal bandwidth in the 5G network of these PoCs, among other functional characteristics of the telecommunications engineering process in real time.

### 4.2. Proof of Concept 1

We performed the PoC-1-1 with a verification test of the latency and bandwidth of the ultrasound exam, by capturing files over the 5G network from Philips portable equipment, model Lumity, type C5-2 Transducer. Its technical specifications are as follows:Model Limity C5-2 transducer;Curved broadband;Operating frequency range from 2 to 5 MHz;Curvature radius of 50 mm;2D color Doppler imaging, M-mode, advanced XRES for multivariate harmonic imaging, SonoCT;High-resolution diagnostic imaging for deeper applications: preconfigured enhancements for abdominal, gallbladder, gynecology/obstetrics and lung imaging;Central venous catheter marker;USB-C Transducer.

We performed PoC-1-2 with a latency verification test and ultrasound scan bandwidth, by capturing the files over the 5G network from Mobisson portable equipment, model M3. Its technical specifications are as follows:Transducer: Model M3;Convex;Weight: 240 g;Wireless;Operating Frequency Range from 3.5 to 5 MHz;Depth from 90 to 280 mm;3-h autonomy in continuous use, 6-hour in standby;Features: Harmonics, Sector Gain, Noise Smoothing;Applications: Performs abdominal and cardiac ultrasound, obstetrical and prenatal exams, renal evaluations and quick identification of lesions and internal bleeding.

#### Performing Ultrasound Exams on PoC-1-1 and PoC-1-2

The PoC-1-1 tests were performed with the Philips portable ultrasound examination equipment. Figure 16 shows the patient lying on a stretcher, with the USG procedure in the abdomen being performed by Dr. 1, an interventional and endovascular surgery radiologist at HCFMUSP.

The other professional, Dr. 2, an interventional radiologist, handled the screens for analysis of the ultrasound of the patient lying on the stretcher at a distance, analyzing the cameras where the patient was in the process of telemedicine. He observed and analyzed the test process with Dr. 1, conducting real-time guidance on the distance ultrasound, in addition to capturing the images to store them, and later carry out the clinical report of the exam.

That is, the focus was on the medical assessment of the equipment use case by Dr. 2. He subjectively assessed whether, with the proposed new O-RAN 5G architecture, there will be an improvement or not in the results collected from the images derived from the ultrasound equipment in the PoC-1 tests.

The PoC 1 was carried out on 26 October 2022 on the second floor of the InRad building, where project antenna A is located, and in the radio intervention department where antenna B is located. The 5G signal was generated in the Airspan Core from the datacenter in Itaú located 9 km from HCFMUSP.

In the location of Antenna A (Doctor Central Room), there was a notebook connected to the dedicated CPE to receive the 5G signal dissipated by the antenna installed there. This machine had a headset and a dedicated webcam. In the locality, radiologist Dr. 2 was responsible for receiving the images and directing the other professional with guidelines for conducting the USG transducer. The portable point-of-care USG equipment used for this PoC was the Philips Lumify, which has a dedicated application for transmitting images in real-time, called Reacts.

The proposal to use the Philips-Lumify equipment was because it was the only equipment that had an accompanying streaming application that provided quality, real-time images at the time of PoC-1 so that the guiding physician could offer remote guidance within the scope from the project. Other streaming platforms tested before PoC-1 showed lower image-quality results.

At the location of antenna B, a Samsung T515 tablet was anchored to the 5G network, connected via Wi-Fi 6 to the 5G CPE that receives the signal dissipated in the environment. The Philips-Lumify product application was installed on the tablet, which, when connecting the transducer via cable to the device, enabled the application to view the USG images to be captured. In this locality, Dr. 1 was responsible for performing the exam, interacting with the radiology doctor in the orientation room.

The radiological images of the ultrasound exam were transmitted using the Reacts Platform, integrated into the Philips-Lumify device application, which required registering two accounts in the Reacts application, executor and advisor. The PoC was divided into the stage of perception of executing professionals and USG exam advisor in real-time in relation to the exam, as well as the technical verification by the application analyst in relation to data traffic in the 5G network.

When the examiner viewed the images, the product was verified to deliver the versioning of the interface and functionality expected by the project, according to the perception of the physician who performed this role.

When transmitting the images to the advisor, the exam executor again expressed that the visualization interface for the exam was satisfactory by dividing his screen into three blocks: one in the upper left corner with his image from the front camera of the tablet, another with the advisor’s image in the upper right corner, and occupying the remaining 3/4 of the screen with USG images and application functionalities.

This stage therefore follows as expected for the project. When connecting to the Reacts platform, the application demonstrated good responsiveness, managing to transmit the images to the guiding medical professional located at the other end.

It was possible to clearly hear the communication between the two professionals; the images were transmitted with minor limitations, possibly related to the network connection speed and limitations of the tablet, which was still connected to the 5G CPE Wi-Fi network. The examination advisor in the report room questioned some difficulty in receiving the images on the Reacts platform, possibly due to connectivity and to the device used to run the application, the Samsung T515 tablet.

The advisor reported that it was possible to direct the information to the executor to carry out the instructions for viewing the ultrasound images on the screen, but with limitations. In general, the advisor reported that this moment was acceptable for the first version of the PoC. However, he showed that he was not satisfied with using only one solution and he hoped that the PoC would be carried out for other USG devices, such as the brand Mobissom, connected to Reacts or another similar telemedicine application.

### 4.3. Proof of Concept 2

We performed the PoC-2 with a set of image file exams (*DICOM) generated with MRI/CT on a patient at the Intervention Center of the InRad of HCFMUSP, as illustrated in Figure 17. The MRI/CT (Magnetic Resonance Imaging/Computed Tomography) from Siemens Healthcare Scanner Room with Softwares sVC (syngo Virtual Cockpit Modality and Steering Clients type VA16A) remote access is a multimodality reading solution built on a client–server platform. In legacy solutions, the machine technician is usually located in a room adjacent to the MRI/CT. sVC allows an operator to remotely command the CT machine from a different location. With this, hospitals can benefit from operational costs and specialized doctors can be more readily available.

All MRI/CT image files were sent and obtained via PACS (Picture Archiving and Communication System) via sVC software (syngo Virtual Cockpit Modality and Steering clients) to facilitate handling and visualization via 2D and 3D in the 5G private network.

In PoC-2-1, we monitor, manipulate and visualize images from the MRI/CT unit by analyzing the latency of packet times from one point to another in the network and the bandwidth that defines the speed and capacity of the 5G private network, as shown in Figure 18.

Figure 19 presents the architecture of the 5G network for Modality/sVC simulation.

### 4.4. Results and Discussion

The tests carried out for the use cases of data received and sent from exams, images, voice and text to evaluate the data transmission capacity of 5G connectivity demonstrated the effectiveness of our methodology.

PoC-1 evaluated the connectivity of the 5G network in terms of transmitting USG images in a product already commercially available that benefits from a dedicated communication platform, Reacts. Hence, with a specific product that offers the opportunity for collaboration, the project team can have answers to questions that are still unanswered. In fact, the project aims to cover a large number of portable USG equipment to perform USG exams in remote locations. In view of the above, the scenario proposed for the examination follows the expected scope, and we collectively chose to consider the PoC satisfactory, due to the limitations found.

Figure 20 presents the results of the second phase of the 5G private network test for PoC-2-1, which are also reported and discussed in Table 2.

## 5. Conclusions

OpenCare5G is a pioneering project in Brazil that is already contributing to quality healthcare through the InovaHC of HCFMUSP. In addition to the implementation of 5G this year, the OpenCare5G project innovates and aims to attract investors from various sectors of information technology, telecommunications and other industries to promote further research in different areas of medicine and engineering as well as foster a complete technology ecosystem, through medtechs, healthtechs and market partners.

The formation of an ecosystem with market partners facilitates this strategy and in this first phase, the 5G private network at HCFMUSP is active and carrying out PoC tests with portable ultrasound equipment and computational tomography via the 5G Network. InRad wants the technology to allow an operator, who can be a healthcare professional, a nurse or a paramedic, to perform the tests remotely with the help of a specialist doctor at the other end, who will also evaluate the images.

The adherence of the 5G network to the demands and tests carried out in this work is evident. We are not focusing on more complex scientific–technological issues such as remote surgeries at this time. Preliminarily, we aim to evolve the state of the art to expand and democratize 5G access to improve quality of life, causing social and economic impacts.

Any network intermittency may have occurred (e.g., some process in the DU/CU consuming too much memory or CPU) in our tests. Some process within the Siemens solution is consuming a lot of memory/processing within the personal computers involved in the tests. We are still investigating the observed latency increase.

Applications of the mobile network for exams do not have issues such as sensitivity to latency in critical missions, as in the case of robotic surgeries. The scope is connectivity that allows images to be sent for follow-up in near-real time. The analysis of a diagnosis currently takes days. This fact is thus important for society to understand that there are several cases of 5G/B5G use. Robotic surgery is a future goal, but this little investment in connectivity will already make an enormous difference in the future.

### Future Work

The OpenCare5G project has a forward-looking expansion plan that includes:Urban PoC: Expanding the concept demonstrated in PoC-1-1-1 to an urban environment within the State of São Paulo. The ultrasound exam room will be located in a basic healthcare unit in the range of 200 km to 400 km from the central room doctor based at InRad of HCFMUSP. The remote ultrasound room will constitute an additional node to the private 5G network, deploying a local DU/Fronthaul switch infrastructure at the UBS. The medical procedure will be refined and submitted for the approval of the relevant medical committees to evaluate actual patients of the public healthcare system. We aim to deploy this phase within one year of finishing the current PoCs;Rural PoC: Expanding the concept demonstrated in PoC-1-1-1 to a remote environment within the Brazilian Amazon region. The ultrasound exam room will be located in an Indigenous basic healthcare unit or a similar location and a reference central room doctor will be deployed in the city of Santarém, State of Pará. Alternatives to backhaul coverage, such as low-orbit satellites, may be considered. The medical procedure will be refined and submitted for the approval of the relevant medical committees to best cater to the native population in the region. We aim to deploy this phase within one year of finishing the current PoCs;Geographical expansion: Expanding the urban and rural PoCs to additional regions in the country;Functional use cases: Increasing the usage of ultrasound diagnosis in prenatal care, heart exams and calculus detection, among others. Addition of other remote assistances to the portfolio. Implementation of further sVC (remote operation for tomography and magnetic resonance machines) in 5G-connected hospitals;Implementation of AI image analysis;Promotion of medtechs/healthtechs’ engagement and knowledge diffusion;Evaluation of synergistic technologies in “green or ecologically correct networks to support the expansion of the intelligent connected edge with minimal network energy consumption;Implementations for network security to overcome challenges in patients’ data privacy and confidentiality;Advanced testbeds for novel communication technologies, including network slicing [62,63,64].

## Figures and Tables

**Figure 1 sensors-23-01047-f001:**
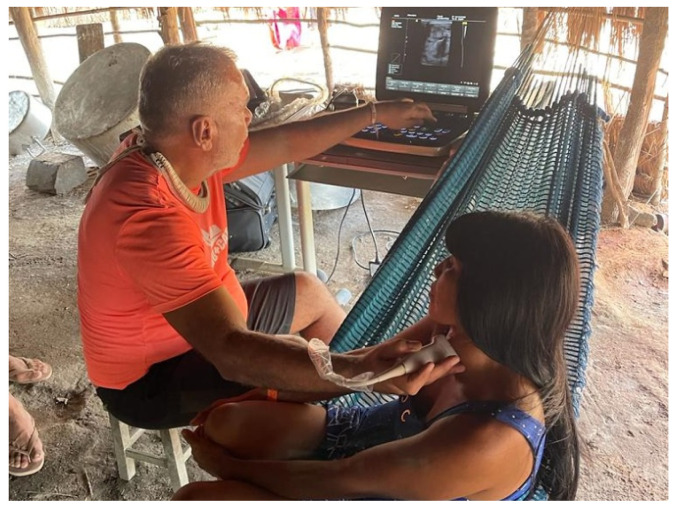
Indigenous patient care by InradHC doctors in the village of Alto do Xingu.

**Figure 2 sensors-23-01047-f002:**
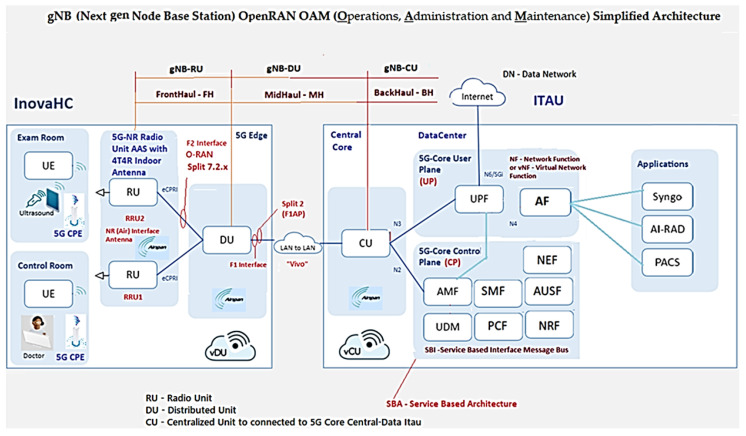
5G xHaul network architecture of the Inovac OpenCare5G Project with simplified O-RAN OAM in the infrastructure layer. Based on [21].

**Figure 3 sensors-23-01047-f003:**
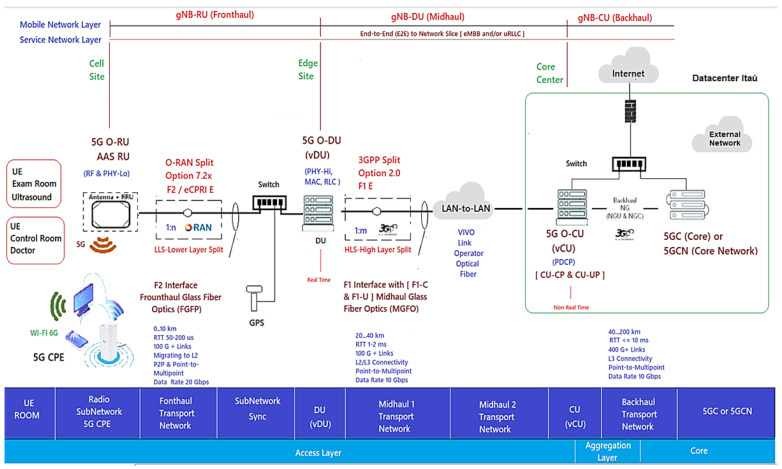
Architecture of the 5G xHaul network of the Inovac OpenCare5G with O-RAN OAM, complete vRAN in the infrastructure, mobile and services layers. Based on [14,26,27,28].

**Figure 4 sensors-23-01047-f004:**
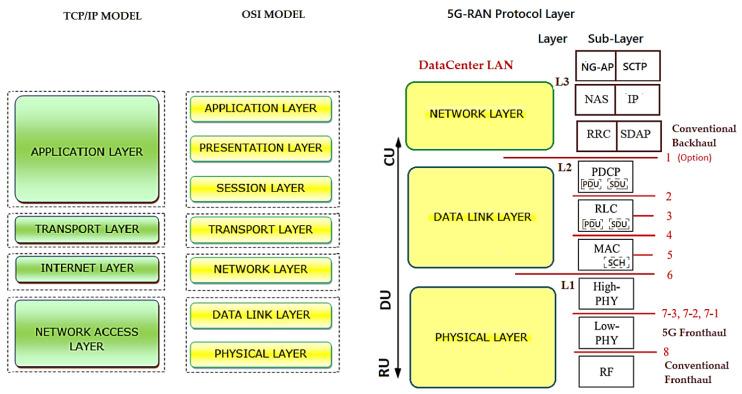
Radio protocol stack simplified in the UMTS/LTE/5G air interface or datacenter LAN with 4G/LTE/5G RAN Network stacks—Comparison to have the “5G NR/NGC SA Protocol Stack”. Based on [32,33,34,35,36].

**Figure 5 sensors-23-01047-f005:**
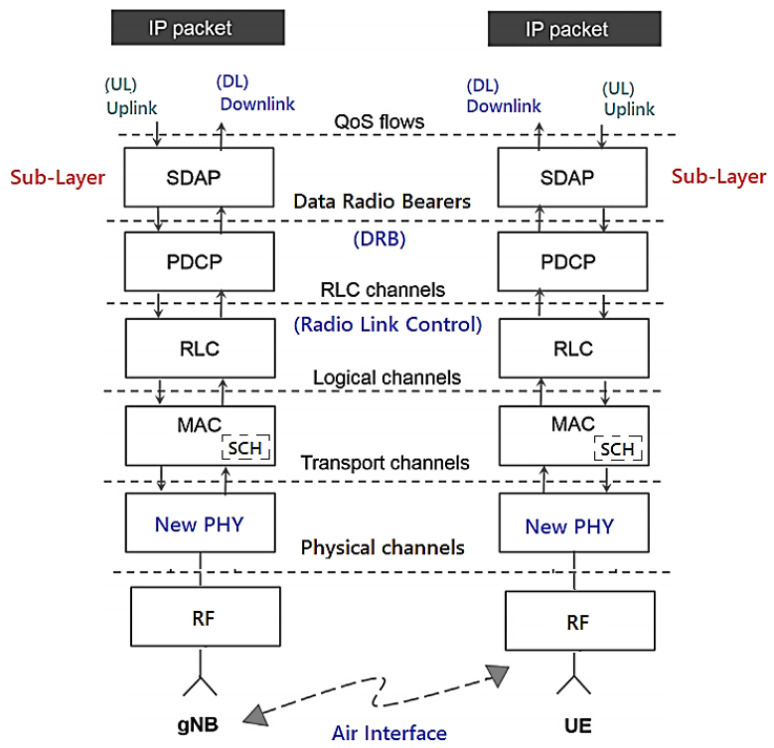
5G NR/NGC SA protocol stack. Based on [38,39].

**Figure 7 sensors-23-01047-f007:**
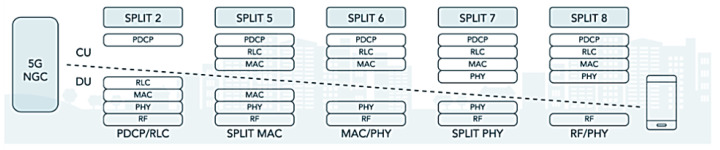
RAN splits—logical to sub-Layer 5G NR/NGC SA. Based on [46].

**Figure 8 sensors-23-01047-f008:**
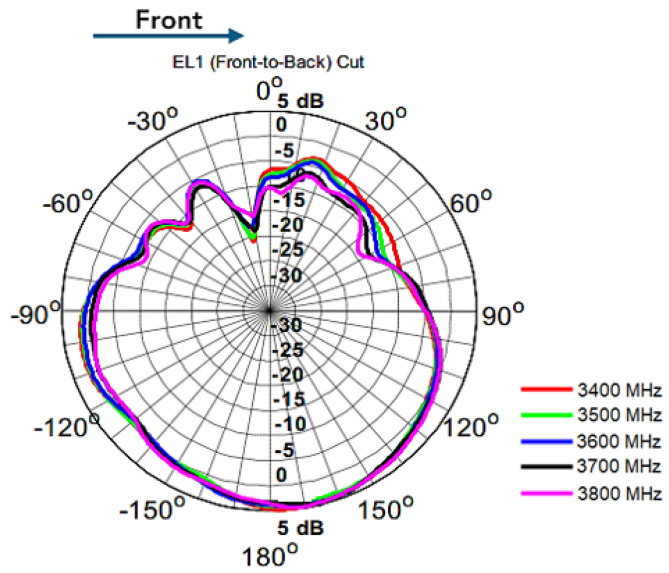
Antenna radiation diagram. Based on [48].

**Figure 9 sensors-23-01047-f009:**
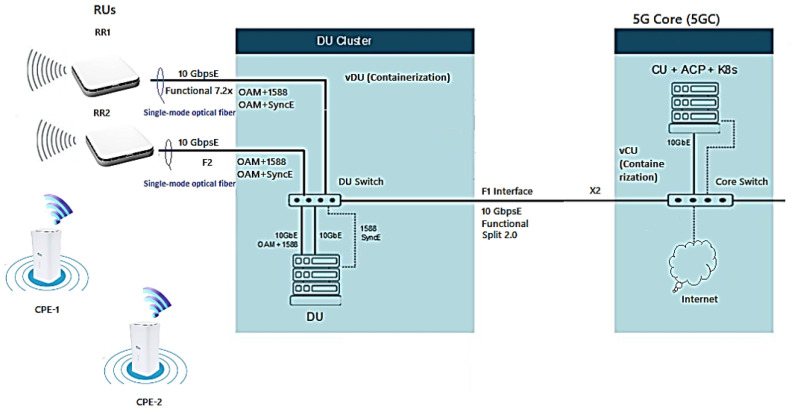
Simplified architecture of RUs+CPEs+DU+5GC. Based on [28,50].

**Figure 10 sensors-23-01047-f010:**
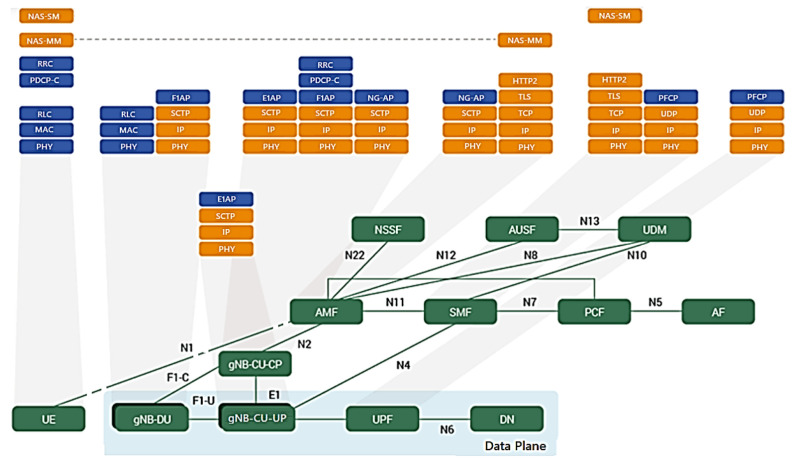
Architecture of the core 5G SBA and reference point with the end-to-end control plane protocol stack and its main interfaces. Based on [51].

**Figure 11 sensors-23-01047-f011:**
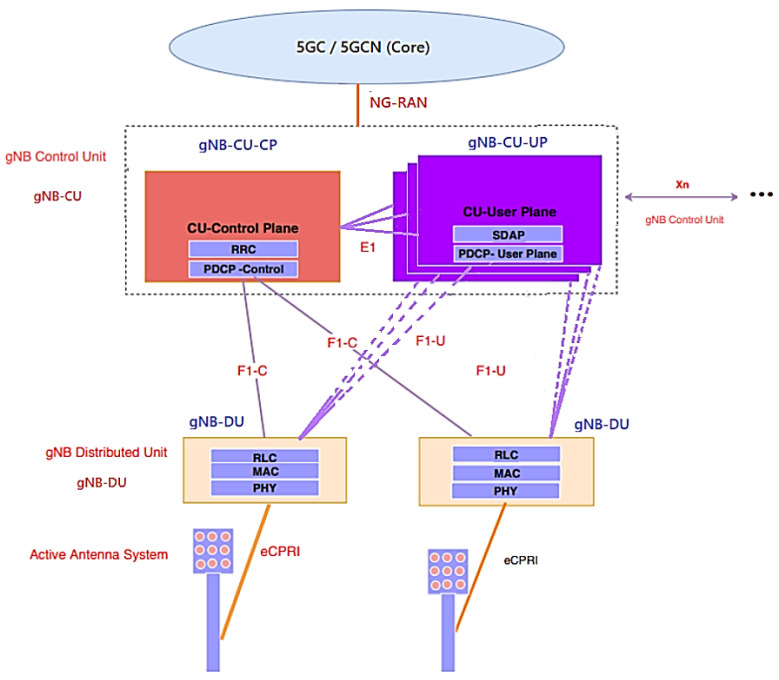
gNB-CU-CP and gNB-CU-UP separation architecture. Based on [50,52,53,54].

**Figure 12 sensors-23-01047-f012:**
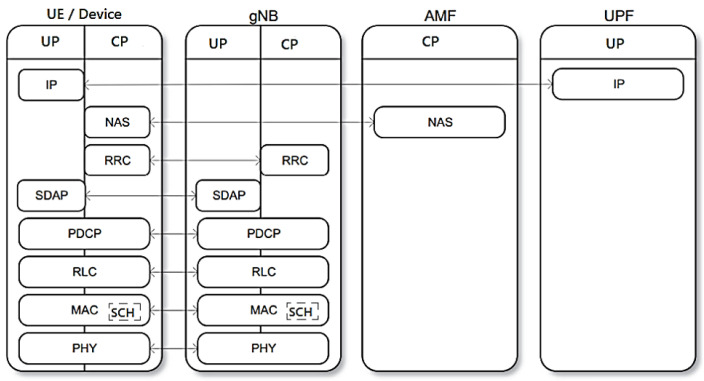
Access stratum for the user and control planes for UE/device and gNB protocol stack with network functions AMF and UPF. Based on [37,40,42,52].

**Figure 13 sensors-23-01047-f013:**
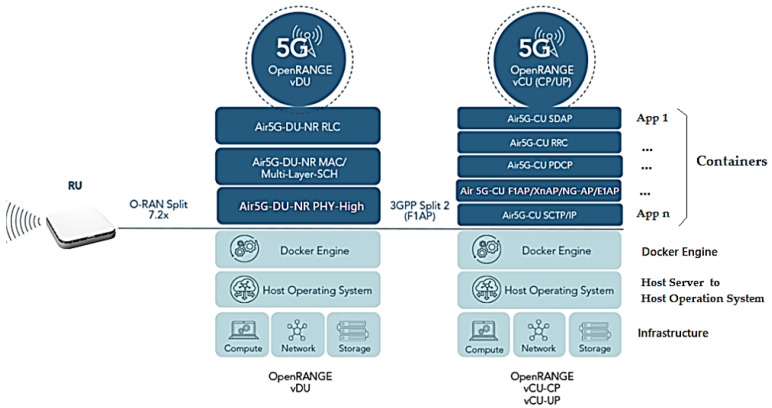
Architecture of Container Network Functions (CNF) with applications running on O-DU/O-CU Servers with OpenRange Software for vDU and vCU. Based on [46].

**Figure 14 sensors-23-01047-f014:**
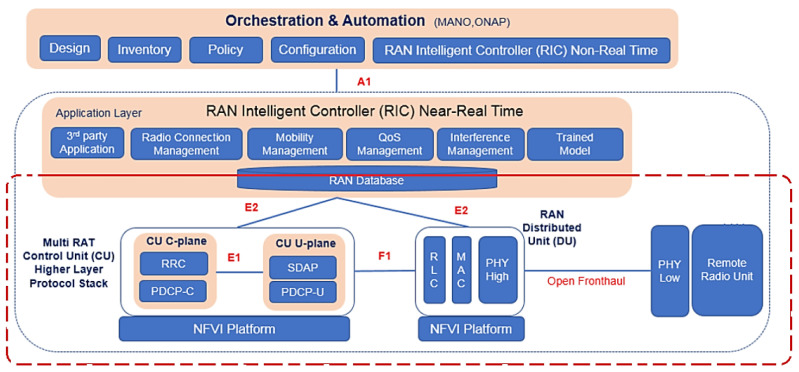
Architecture of CNF with applications running on O-DU/O-CU servers with OpenRange Software for vDU and vCU. Based on [58,59].

**Figure 15 sensors-23-01047-f015:**
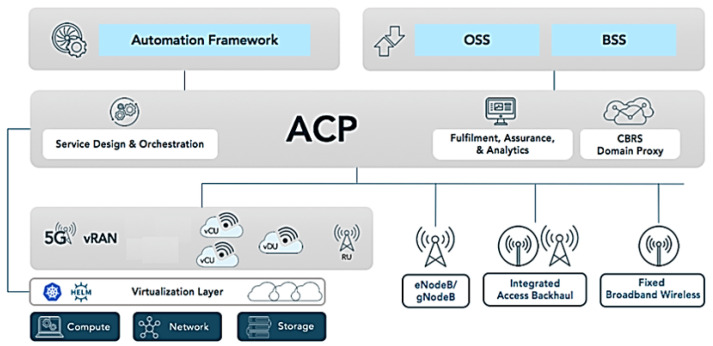
ACP—Airspan Control Platform. Based on [60].

**Figure 16 sensors-23-01047-f016:**
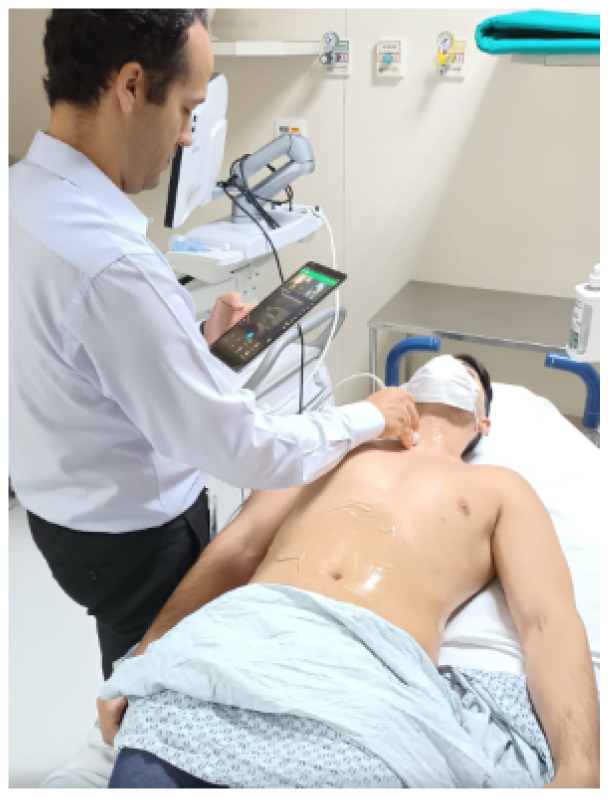
Doctor 1 performing ultrasound on the patient.

**Figure 17 sensors-23-01047-f017:**
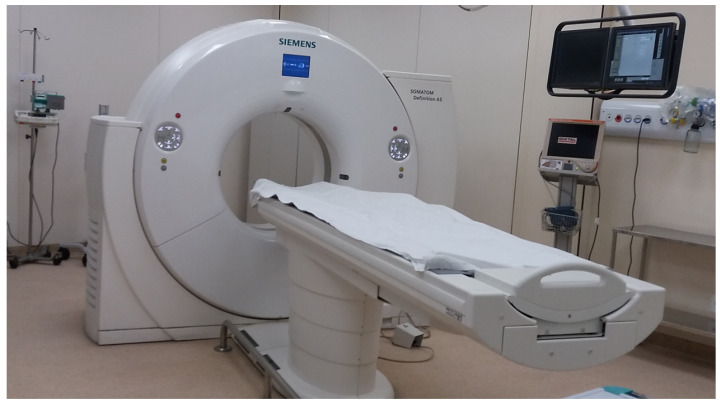
MRI/CT scanner room.

**Figure 18 sensors-23-01047-f018:**
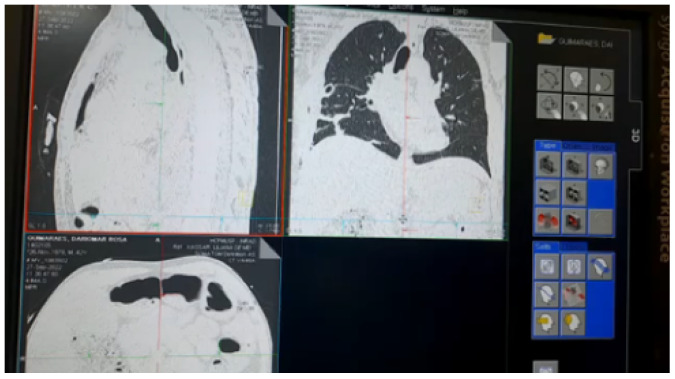
Manipulation and visualization of images from the tomograph via 5G network.

**Figure 19 sensors-23-01047-f019:**
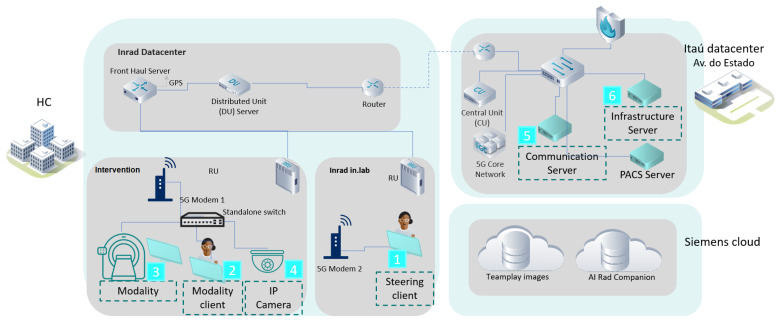
Architecture with the 5G network for Modality/sVC Simulation.

**Figure 20 sensors-23-01047-f020:**
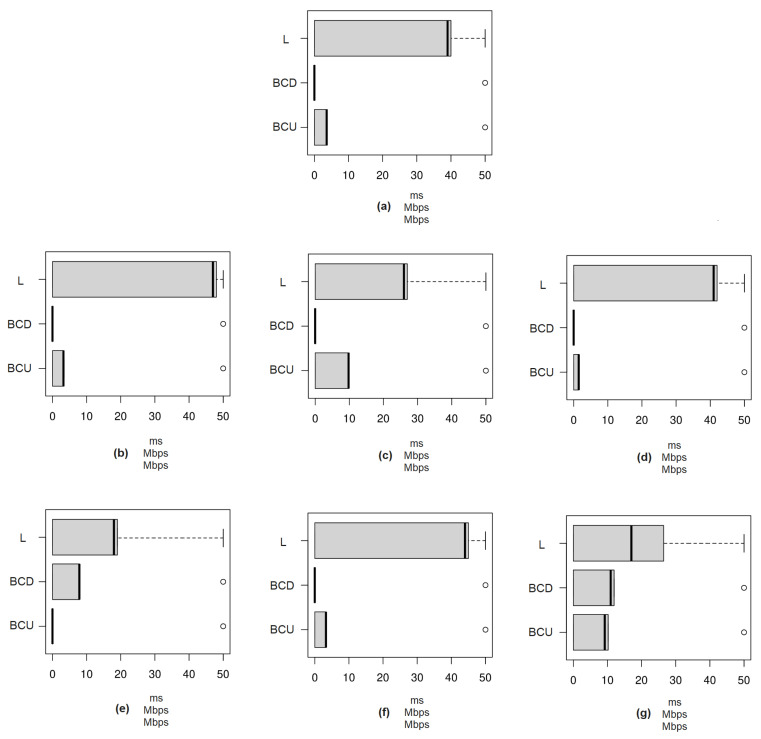
Boxplot of tests in the second phase of the 5G private network for PoC-2-1. (**a**) Test 1: VNC; (**b**) Test 2-1: VNC; (**c**) Test 2-2: PACS; (**d**) Test 3-1: VNC; (**e**) Test 3-2: PACS; (**f**) Test 4-1: VNC; (**g**) Test 4-2: PACS. BCD: Bandwidth Consumption in Download; BCU: Bandwidth Consumption in Upload; L: Latency.

**Table 1 sensors-23-01047-t001:** Summary of related work.

Work	Contribution	Limitations	Future Work
Singh, Singh and Kumbhani (2020) [15]	The disaggregation of hardware and software creates a unified architecture through various advancements and brings many benefits such as low latency and network slicing. In addition to facilitating network automation, O-RAN offers several benefits such as agility, deployment flexibility, real-time responsiveness and operating cost reduction.	Reduction in operating cost by estimating the plug-and-play feature of O-RAN; modern learning methods can reduce maintenance cost up to 80%.	Implementing RIC policies close to real-time and non-real-time control loop meeting economic and ecological aspects; coordination, updating and training are difficult with modern ML and AI learning techniques and the challenging handling of data (between layers).
Wang et al. (2022) [19]	They modeled the RU-DU resource allocation problem in an O-RAN as a packing problem, proposing a self-play reinforcement learning strategy. They applied the combined approach of deep neural network with Monte-Carlo Tree Search (MCTS) to solve this combinatorial optimization problem.	Learning by auto-run eliminates the need for demonstrator data, which can be expensive and time-consuming to collect.	Mapping RU shift requirements to local mobile edge computing centers for future centralized processing that would significantly reduce power consumption in cellular networks. That is, we study the resource allocation problem between the RU-DU of the O-RAN system modeled as a 2D packing problem.
Lekshmi and Ponnekanti (2019) [20]	ML can play significant roles in learning from the wireless environment variations, categorizing problems, anticipating challenges, predicting outcomes, and exploring possible solutions, decisions, and actions.	Current RANs are reactive and base stations run algorithms on the centralized server to meet user demands. However, in 5G, even ms of delay can cause a great impact.	Sustaining advanced RLM requires an efficient implementation of ML-based traffic optimization that can handle large volumes of data in 5G networks.
Polese et al. (2022) [21]	O-RAN networks will dramatically change the design, deployment, and next-generation operations for cellular and other networks, allowing, among other things, transformative applications of ML for RAN optimization and control.	RAN reconfiguration with equipment whose operations cannot be adjusted to support diverse deployments and different traffic profiles; limited co-ordination between network nodes.	Real-time control loops will be included that operate in the real-time domain, i.e., below 10 ms for RRM at the RAN node level, or even below 1 ms for device management and optimization.

**Table 2 sensors-23-01047-t002:** Tests results with sVC in PoC-2-1.

Steering ^1^	Description	Results ^2^	Evaluation
Test 1	Access, via VNC Client on PC1, to the VNC Server on PC2.	[PC2 (Test 1)—VNC protocol: L = 40 ms; BCU = 3.79 Mbps; BCD = NA].	Application responding normally to commands, without loss of performance.
Test 2	Access, via VNC Client on PC1, to the VNC Server on PC2, with image transfer towards PC2 –> PACS Itaú.	[PC1 (Test 2-1)—via VNC: L = 48 ms; BCU = 3.39 Mbps; BCD = NA] / [PC1 (Test 2-2)—via PACS: L = 27 ms; BCU = 9.94 Mbps; BCD = NA].	Application responding normally to commands, without loss of performance.
Test 3	Access, via VNC Client on PC1, to the VNC Server on PC2, with image transfer towards PACS Itaú –> PC2.	[PC1 (Test 3-1)—via VNC: L = 42 ms; BCU = 1.57 Mbps; BCD = NA] / [PC1 (Test 3-2)—via PACS: L = 19 ms; BCU = NA; BCD = 7.93 Mbps].	Application responding normally to commands, without loss of performance.
Test 4	Access, via VNC Client on PC1, to the VNC Server on PC2, with image transfer towards PC2 –> PACS Itaú and PACS Itaú –> PC2.	[PC1 (Test 4-1)—via VNC: L = 45 ms; BCU = 3.38 Mbps; BCD = NA] / [PC1 (Test 4-2)—via PACS: L = 17–26 ms; BCU = 10.21 Mbps; BCD = 11.95 Mbps].	Application responding normally to commands, without loss of performance.

^1^ Requirements proposed for each steering: Network bandwidth ≥ 40 Mbps; Latency ≤ 30 ms. ^2^ BCD: Bandwidth Consumption in Download; BCU: Bandwidth Consumption in Upload; L: Latency; NA: Not Available.

## Data Availability

Not applicable.

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
