# Peer review of "OpenCare5G: O-RAN in Private Network for Digital Health Applications"

_sensors, 2023, doi:10.3390/s23021047_

Round 1

Reviewer 1 Report

In this project the authors introduced the connectivity and capacity of the OpenCare5G Private Network with the Proof of Concepts for transmitting the volume of data from remote exams, using higher speed and lower latency. Further, the authors consider the project as a catalyst for innovation in health, connecting resources and entrepreneurs to generate solutions for the innovation ecosystem of organizations. They started a 5G xHaul O-RAN OAM vRAN Private Network Project Architecture that is able to provide connectivity and interoperability through radios, hardware, software, containers, virtual machines, cloud-native, open interfaces from different manufacturers, and will improve the provision of basic services to populations living in remote areas of Brazil.

My suggestions regarding the improvement of the paper are as follows:

I went through all the text and I did not see any test to minimize energy consumption at each node. I propose if possible to integrate any kind of Energy Aware Software to consider the energy consumption for network as a whole.

What about Network Security, do you plan to implement it in future in order to protect the patient’s sensitive data?

The field tests carried out with “real use cases” of data received and sent from exams, images, and voice to evaluate the data transmission capacity of 5G connectivity,  seem to demonstrate the effectiveness of used methodology.

The findings are adequate and contribute with several proposals, tests and solutions in the area of quality healthcare.

The proposed 5G xHaul Network Architectures are added values of the project.

Author Response

Dear Reviewer 1,

Reviewer 2 Report

-5.1. Future Work must be better presented. It is better to have a good paragraph for the future works. 

-Appropriate references must be considered in the introduction section. 

-Research implication and method applications must be presented in the discussion section.

Author Response

Dear Reviewer 2,

Reviewer 3 Report

This is a well-written text on a new and interesting contemporary scientific topic which has important theoretical and clinical implications. However, I find that its content does not address fundamental issues of 5G and e-Health linked to critical security and privacy issues of the OpenCare5G pioneering project. I believe that the text will be significantly improved by the addition of information connected with security threats and challenges on privacy, confidentiality, safety, well-being, and fundamental freedoms.

Author Response

Dear Reviewer 3,

Round 2

Reviewer 2 Report

The paper  has been improved and can be published by Sensors.